# Construction of a MoO_x_/MoS_2_ Heterojunction via the Surface Sulfurization of the Oxide and Its Photocurrent-Switching Characteristics in the Range of the Broadband Light Spectrum

**DOI:** 10.3390/ma17225507

**Published:** 2024-11-12

**Authors:** Xingfa Ma, Xintao Zhang, Mingjun Gao, You Wang, Guang Li

**Affiliations:** 1School of Environmental and Material Engineering, Center of Advanced Functional Materials, Yantai University, Yantai 264005, China; zhangxintao@ytu.edu.cn (X.Z.); gaomj@ytu.edu.cn (M.G.); 2National Laboratory of Industrial Control Technology, Institute of Cyber-Systems and Control, Zhejiang University, Hangzhou 310027, China; king_wy@zju.edu.cn (Y.W.); guangli@zju.edu.cn (G.L.)

**Keywords:** MoO_3_ nanosheets, surface S/O exchange of oxide, MoO_3_/MoS_2_ surface heterojunction, photocurrent-extracting, optoelectronic response in broadband light spectrum

## Abstract

In order to utilize the longer wavelength light, the surface sulfurization of MoO_3_ was carried out. The photocurrent responses to typical 650, 808, 980, and 1064 nm light sources with Au gap electrodes were investigated. The results showed that the surface S–O exchange of MoO_3_ improved the interfacial charge transfer in the range of the broadband light spectrum. The S and O can be exchanged on the surface of MoO_3_ nanosheets under the hydrothermal condition, leading to the formation of a surface MoO_x_/MoS_2_ heterojunction. The interfacial interaction between the MoO_3_ nanosheets and MoS_2_ easily generated free electrons and holes, and it effectively avoided the recombination of photogenerated carriers. Meanwhile, the surface S-doping of MoO_3_ also resulted in the generation of an oxygen vacancy and sulfur vacancy on MoO_3−x_S_2−y_. The plasmonic characteristics of MoO_3−x_ contributed to the enhancement of the interfacial charge transfer by photoexcitation. Otherwise, even with zero bias applied, a good photoelectric signal was still obtained with polyimide film substrates and carbon electrodes. This indicates that the formation of the heterojunction generates a strong built-in electric field that drives the photogenerated carrier transport, which can be self-powered. This study provides a simple and low-cost method for the surface functionalization of some metal oxides with a wide bandgap.

## 1. Introduction

Layered two-dimensional functional metal oxides and their nanocomposites have received considerable attention in terms of new energy [1,2,3,4,5,6], energy storage, H_2_ production [7,8,9,10,11,12,13,14], supercapacitors [15], light-emitting devices [16], photodetectors [17,18,19], memristors [20], thermoelectric devices [21], electrochromic devices [22], photonics, electronics, artificial synapse, chemical sensors [23], humidity sensors [24], biosensors, photocatalytic fields [25,26,27], optoelectronics, excitonic devices, valleytronics, and so on. While transition metal chalcogenides (TMDCs) are one kind of vdW two-dimensional material with a typical bandgap in the range of 1–2 eV, they have attracted tremendous interest in relation to light-active materials and optoelectronics fields, especially in the near-infrared region. Among them, MoS_2_ is one of the typical TMDCs, which undergoes a transition from an indirect bandgap (about 1.2 eV) in its bulk form to a direct bandgap (about 1.8 eV) in its monolayer. In general, the precursors used to synthesize MoS_2_ are Mo or MoO_3_, which can be sulfurized to MoS_2_ by controlling the chemical reaction conditions. Conversely, MoS_2_ can also be oxidized to MoO_3_ by changing the reaction parameters. Therefore, MoO_3_ and MoS_2_ can be interconverted to modify the photophysical properties of the material in the visible and the near-infrared regions by controlling the process technology and surface treatment factors. On the other hand, MoO_3_ is an n-type semiconductor with a wide band gap (about 3.1 eV), which has also been widely used in storage materials, gas sensors, photochromism devices, electrochromism devices and catalysis fields. Compared with other semiconductors, the remarkable features of MoO_3_ are its high work function, high carrier mobility and excellent stability. Therefore, MoO_3_ is also a good candidate for hole-transporting materials in electronic or optoelectronic devices. The MoO_3_/MoS_2_ heterostructure combining the wide and narrow bandgap materials would have synergistic or complementary effects, and it would have potential applications in interdisciplinary fields. Therefore, it is a simple and effective way to construct an MoO_3_/MoS_2_ heterojunction by using MoO_3_ as a matrix.

Since the properties of materials are strongly dependent on their microstructures, interfaces and defects, doping and defects have significant effects on the photophysical characteristics of MoO_3_. To date, the research focus on MoO_3_-based nanocomposites is still concentrated on elemental doping, oxygen vacancy modulation, band gap engineering, surface functionalization or surface reconstruction, and the heterostructure. To give an overview of the main research in these areas, some representative research examples are shown below. Yoon and co-workers [28] constructed the van der Waals heterostructure of atomically layered α-MoO_3_ on MoS_2_ by the oxidation approach. Giannazzo and co-workers [29] tailored the Schottky barrier height of MoS_2_ by oxygen plasma functionalization. Reidy and co-workers [30] revealed the atomic-scale mechanisms of MoS_2_ oxidation for kinetic control of MoS_2_/MoO_3_ interfaces. Roy and co-workers [31] studied the oxygen-vacancy-induced band engineering of MoS_2_/MoO_3_ and applied it to confined catalysis. Pondick and co-workers [32] carried out the stepwise sulfurization of MoO_3_ to MoS_2_ by chemical vapor deposition. Hong and co-workers [33,34] synthesized MoS_2_ layers by direct sulfidation of MoO_3_ surfaces. Patra and co-workers [35] carried out the oxidation of MoS_2_/GO to MoS_2_/MoO_3−x_/RGO and obtained plasmonic 2D multifunctional nanocomposites for solar hydrogen generation from the near-infrared and visible regions. Li and co-workers [36] reported the light-induced in situ formation of a non-metallic plasmonic MoS_2_/MoO_3−x_ heterostructure with efficient charge transfer for CO_2_ reduction and SERS detection. Fatima and co-workers [37] performed a comparative study between molybdenum sulfurized MoS_2_ and molybdenum trioxide precursors for thin-film device applications. Romanov and co-workers [38] synthesized large-area two-dimensional MoS_2_ films by sulfurization of atomic-layer-deposited MoO_3_ thin films for nanoelectronic applications. Karmakar and co-workers [39] studied the photo-induced exciton dynamics and achieved broadband light harvesting in ZnO nanorod-templated multilayered two-dimensional MoS_2_/MoO_3_ photoanodes for solar fuel generation. Xiao and co-workers [40] fabricated nano-caved MoS_2_/MoO_2_ hybrids. Zhu and co-workers [41] carried out remote plasma oxidation and atomic layer etching of MoS_2_. Weber and co-workers [42] studied the basic reaction steps in the sulfidation of crystalline MoO_3_ to MoS_2_ by X-ray photoelectron and infrared emission spectroscopy. Park and co-workers [43] synthesized vertical MoO_2_/MoS_2_ core–shell structures on an amorphous substrate by chemical vapor deposition. Khondaker and co-workers [44] studied the effects of plasma treatment on the bandgap of MoS_2_ and investigated the layer-dependent relationship. Jung and co-workers [45] studied the nucleation and growth of monolayer MoS_2_ in multi-steps of MoO_2_ crystals by sulfurization. Nualchimplee and co-workers [46] reported the auto-oxidation of exfoliated MoS_2_ in N-methyl-2-pyrrolidone. Kumar and co-workers [47] observed the positive trions in α-MoO_3_/MoS_2_ van der Waals heterostructures. Liu and co-workers [48] enhanced the hydrodeoxygenation of lactic acid with the synergistic interaction between the vacancy and thermoelectric properties in an MoS_2_/MoO_3_ composite. Shahrokhi and co-workers [49] tailored the optoelectronic and dielectric properties of few-layer S-doped MoO_3_ and O-doped MoS_2_ bulk nanosheets. Shahrokhi and co-workers [50] discussed the understanding of the optoelectronic properties of S-doped MoO_3_ and O-doped MoS_2_ bulk systems. Phalswal and co-workers [51] synthesized layered MoS_2_ nanostructures and then transformed them into MoO_3_ nanoparticles using a microwave. Ftouhi and co-workers [52] synthesized MoS_2_ and MoO_3_/MoS_2_ hybrid thin films using a sulfidation technique. Pushpalatha and co-workers [53] modulated the crystallinity of 1D MoO_3_ and its conversion to 2D MoS_2_ nanosheets for efficient hydrogen evolution. Pal and co-workers [54] reported Si-compatible MoO_3_/MoS_2_ core–shell quantum dots for wavelength-tunable photodetection over a wide visible range. Liu and co-workers [55] synthesized interlayer expanded MoS_2_ by sulfurization of MoO_3_ with enhanced sodium-ion storage. Shaji and co-workers [56] reported the stepwise sulfurization process of MoO_3_ to MoS_2_ thin films, which was studied by real-time X-ray scattering. Liu and co-workers [57] studied the morphological and structural evolution of α-MoO_3_ single crystal belts toward MoS_2_/MoO_2_ heterostructures during post-growth thermal steam sulfurization. Yang and co-workers [58] carried out the morphology engineering of MoS_2_ nanostructures by controlling the MoO_3−x_ concentration. Choi and co-workers [59] investigated the effects of the plasma conditions on the sulfurization of MoO_3_ thin films and surface evolution for the formation of MoS_2_. Bortoti and co-workers [60] reported a facile and inexpensive oxidative conversion of MoS_2_ into α-MoO_3_, focusing on the synthesis, characterization, and application. Španková and co-workers [61] investigated the influence of the precursor thin-film quality on the structural properties of large-area MoS_2_ films grown by sulfurization of MoO_3_ on sapphire. Mouloua and co-workers [62] prepared an MoS_2_/MoO_2_ nanocomposite by chemical vapor deposition for optoelectronic applications. Kumar and co-workers [63] investigated the role of different environments in the sulfurization of MoO_3_ to MoS_2_. Zhang and co-workers [64] developed surface partially oxidized MoS_2_ nanosheets as a more efficient cocatalyst for photocatalytic hydrogen production. Saleem and co-workers [65] studied electrocatalytic hydrogen evolution on sulfur-deficient MoS_2_ nanostructures. Zhou and co-workers [66] reported plasmonic oxygen defects in MO_3−x_ (M = W or Mo) nanomaterials, focusing on the synthesis, modification, and biomedical applications. Wang and co-workers [67] studied the vacancy defects in 2D transition metal dichalcogenide electrocatalysts, focusing on the aggregate and atomic configuration. Liu and co-workers [68] reported a two-dimensional amorphous plasmonic heterostructure of Pd/MoO_3−x_ for enhanced photoelectrochemical water splitting performance and so on.

In reviewing the recent references to MnO_2_/MoS_2_ in the above-mentioned major research areas, it is found that doping, surface, interface, and defect engineering, especially vacancy defects such as oxygen vacancy and sulfur vacancy, are the key approaches to improve the properties of materials and expand their interdisciplinary applications. Most of the synthesis of MnO_2_/MoS_2_ includes chemical vapor deposition, sulfurization of MoO_3_, oxidation of MoS_2_, plasma treatment, S-doped MoO_3_, and O-doped MoS_2_ with different approaches. It can be seen that oxygen and sulfur can be exchanged by controlling the chemical reaction parameters. Oxygen and sulfur doping is an important means of MoS_2_ or MoO_3_ functionalization. In previous publications, the focus of the Ma group has been on studying the examination of the gas sensitivity of organic/inorganic hybrids. Since a large number of nanocomposites exhibit multi-functionalities, different physical mechanisms coexist. In the last decade, the work of the Ma group [69,70] has transferred to the defect-assisted photoconductive behaviors of nanocomposites. The focus of the research is on the contribution of the interfacial interaction of nanocomposites to the enhancement of the photophysical properties. Ma and co-workers [71] studied the interfacial electronic interaction between MoO_3_ and polymer-derived carbon nanomaterials and improved the extraction capability of photogenerated carriers. Doping would create more new interfaces and defects and lead to significant changes in the photophysical properties of materials through interfacial contact and the formation or distribution of defects. Defect engineering of materials can effectively control the different physical mechanisms, photophysical properties, and interdisciplinary applications. The physical mechanism of photoconductive responses mainly depends on the charge trap-assisted (such as electron-trap or hole-trap) photoconductive behavior. Therefore, the photoelectric performance can be enhanced by the defect engineering of nanocomposites.

In this work, in order to extend the light harvesting in the visible and NIR region and to improve the extraction ability of carriers of MoO_3_ heterojunctions by light induction, the surface S-doping of MoO_3_ was carried out utilizing a two-step hydrothermal treatment. The prefabrication of MoO_3_ was carried out by the hydrothermal method, and then the surface partial sulfurization of MoO_3_ was performed in the presence of sulfur resources. The electron/hole separation and extraction abilities of the MoO_3−x_S_y_ nanojunctions were investigated by weak visible and NIR light induction. It showed that the MoO_3−x_S_y_ nanosheets exhibited obvious optoelectronic characteristics in a wide range of light spectra. This method is also suitable for surface modification of similar material systems, such as WO_x_S_y_. Based on the facile and inexpensive synthesis of heterostructure or surface functionalization, these results are also helpful for surface modification of some oxides with a wide bandgap. It is helpful to explore the photodynamic and electronic interaction of light/matter for different material systems via surface doping modification of oxide materials. On the other hand, there are many main research directions in photoactive materials. We do not intend to follow these fixed research directions but only to provide some common, complementary content from an interdisciplinary perspective. Otherwise, the interfacial contact is very important for the construction of heterojunctions. There are also many ways of constructing heterojunctions, most of which involve growing a second material in the presence of one. The interfacial defect is unfavorable for charge transfer. Surface doping of wide bandgap oxides can improve the charge transport and avoid the interfacial gap. Using the same material to construct a heterojunction by surface doping can also avoid the lattice mismatch caused by two materials and improve the interfacial contact, which is beneficial for photogenerated carrier transport.

## 2. Experimental Part

### 2.1. Raw Materials

Sodium molybdate (Analytical Reagent, AR), Tianjin Fengchuan Chemical Reagent Co. (Tianjin, China). Hydrochloric acid (AR), Hongteng Weiye New Material Co., Ltd. (Yantai, China). Thiourea (AR), Tianjin Taixing Reagent Factory (Tianjin, China).

### 2.2. Preparation of MoO_3_

The preparation of diluted hydrochloric acid and MoO_3_ is described in reference [71]. The hydrothermal reaction condition was held at 170 °C for 6 h.

### 2.3. Preparation of MoO_3−x_S_y_

The first step was the same as in Section 2.2 (synthesis of MoO_3_).

After cooling for 24 h, 2 g thiourea was added and stirred for 5–10 min. The hydrothermal synthesis was similar to in reference [71]. The reaction condition was held at 170 °C for 2, 4, and 6 h.

### 2.4. Characterization with SEM, EDS, EDS Mapping, Raman Spectra, 2D Raman Mapping, UV-Vis-NIR Spectrum, and XRD

The details of the characterizations by scanning electron microscope (SEM), UV-VIS-NIR spectrophotometer (UV-Vis-NIR) and X-ray powder diffraction (XRD) were similar to in reference [72]. The instruments used were a ZEISS Gemini SEM300 (Oberkochen, Germany), TU-1810 spectrophotometer (Beijing Puxi General Instrument Co., Ltd., Beijing, China), and XRD-7000 from SHIMADZU (Shimadzu, Kyoto, Japan).

The Raman spectra and 2D Raman mapping were measured using a PHS-3C confocal Raman spectrometer (HORIBA, Kyoto, Japan). The operating wavelength and power density of the laser radiation were 532 nm and 2 mW, respectively.

### 2.5. Photocurrent Response of Nanocomposite to the Visible Light and Part of NIR

The resulting nanosheet suspension was cast on the Au gap electrodes on a flexible polyethylene terephthalate (PET) substrate. The gap distance of the Au gap electrodes was 100 μm. The structure of the Au gap electrodes is shown in Figure 1. The determination of the optoelectronic response to the weak visible light (40 W) or 650 nm (100, 50, and 5 mW) and 808, 980, and 1064 nm NIR (10, 40, 50, 100, and 200 mW) was similar to that in reference [72]. The bias applied was 1 V. The polyimide film substrates and carbon electrodes were also used to study the photoelectric signals over a wide spectral range for heterojunctions with different surface sulfurization times (2, 4, and 6 h). Heterojunctions are formed due to the sulfidation of the oxide surface, and the most important feature of heterojunctions is the formation of built-in electric fields. The built-in electric field can drive the transport of photogenerated carriers. Therefore, some samples were selected to investigate the effect of different bias voltages (0, 1, and 3 V) on the photogenerated current.

## 3. Results and Discussion

There are many ways to modify the surface of materials, including chemical modification (various chemical oxidations, reduction, grafting, coating, doping, etc.), physical treatments (plasma treatment, corona treatment, radiation, plasma grafting, etc.), surface reconstruction, surface functionalization, etc. It is well known that Mo oxides and sulfides can be transformed into each other. Therefore, the surface is partially oxidized or sulfidized by controlling the reaction conditions during the interconversion of Mo oxides and sulfides. Similar surface functionalization has been applied to Mo oxides and sulfides in this study. This modification of the materials is shown in Figure 2.

As shown in Figure 2, it can be seen that surface sulfur doping or oxygen doping can be an important way to functionalize the surface of oxides and sulfides. In this work, the MoO_x_S_y_ nanosheets were obtained by two-step hydrothermal preparation. The first step was the synthesis of the MoO_3_ nanosheets, and the second step was the surface S-doping via O and S exchange near the surface of the MoO_3_ nanosheets under hydrothermal reaction conditions. The representative SEM image of the resulting MoO_x_S_y_ is shown in Figure 1.

As shown in Figure 1, the overall morphology form is a nanosheet structure with a length of about 100 nm. The thickness is about 5–6 nm. These flower-like morphologies should have a high specific surface area due to the absence of aggregation of the nanosheets. It is expected to have excellent adsorption and surface properties. Overall, the uniform morphology of the MoO_x_S_y_ nanosheets is very good. The surface properties of the nanosheets play an important role in their interdisciplinary application.

The XRD results of MoO_x_S_y_ are shown in Figure 2. The UV-Vis-NIR of MoO_x_S_y_ is shown in Figure 3.

As shown in Figure 2, the degree of crystallization of the resulting nanosheets is not very high, while the noise level is higher. However, some diffraction peaks can still be distinguished, such as at 25.8°, 29.4°, 35.27°, and 49.70°, which are the peaks of the (210), (300), (310), and (500) planes of MoO_3_ (PDF# 21-0569), respectively (as shown in Figure 2A). The diffraction peaks at 13.9°, 33.34°, and 59.1° are the peaks of the (002), (101), and (008) planes of 2H-MoS_2_(PDF# 37-1492) (as shown in Figure 2B). Here, the qualitative characterization was introduced in this study. As shown in the latter Raman spectra of the MoO_3_/MoS_2_ surface heterojunction, the MoO_3_/MoS_2_ surface heterojunction with different surface sulfurization times (4 and 6 h) showed clearly the broad characteristic peaks of MoS_2_ and MoO_3._ The presence of oxide/sulfide heterojunctions is further supported by the Raman spectroscopy. The 2D Raman mapping also showed that the characteristic signal peak shifted from molybdenum sulfide to molybdenum oxide when the resulting sample was irradiated with a laser for a long time. This shows that oxides and sulfides can be transformed into each other. Despite the presence of the weak XRD diffraction peaks, several experiments support the formation of oxide/sulfide heterojunctions by the surface sulfidation of the oxides. Therefore, the resulting product contains some MoS_2_ and MoO_3_ components, which form the MoO_3_/MoS_2_ heterojunction. This illustrates that this simple and inexpensive surface treatment is feasible.

Some references also showed some similar weak XRD diffraction peaks. The degree of crystallization of the MoO_3−x_ thin films is also not very high, while the noise level is higher [73].

As shown in Figure 3, the optical absorption of the MoO_x_S_y_ films was found to extend into the NIR. The absorbance in the 700–800 nm region was slightly high. Although this absorption is not very strong, there is a clear absorption peak in this region. Since the bandgap is about 3.1 eV of MoO_3_, the absorption of the broad spectrum is mainly due to sulfur doping. The atomic radius of sulfur is greater than that of oxygen, and the electron-donating capacity of S^2−^ is greater than that of O^2−^. The doping of sulfur not only produces a certain amount of MoS_2_ but also causes some changes in the band gap due to the internal stress created. However, the key to these potential applications depends on the generation of excitons and the separation of free electrons/holes by light excitation of appropriate energy. These factors are strongly dependent on the band gap width and defects of the material. Therefore, the optoelectronic response to the weak visible light has been preliminarily investigated. The representative results are shown in Figure 4.

Figure 4 shows the transient photocurrent response curve of MoO_x_S_y_ to weak visible light. As shown in Figure 4, when the MoO_x_S_y_ film was excited with weak visible light, the current increased dramatically. On the contrary, when the visible light was turned off, the film current decreased significantly. The response time, recovery time, and on/off ratio of MoO_x_S_y_ to weak visible light are shown in Table 1. An obvious photoelectric signal can be obtained. It is expected that the free electrons and holes could be involved in a number of chemical reactions of reduction and oxidation or physical signal extraction in interdisciplinary applications.

In the visible light range (400–700 nm), 650 nm is an important light resource, widely used for the artificial retina, simulated photosynthesis, and photoelectric imaging, etc. This light is also used in solar cells, photocatalysis fields, and photodetectors. Therefore, in the visible light region, the optoelectronic responses of MoO_x_S_y_ to 100 mW 650 nm were investigated in this study because 650 nm is close to 700 nm. The results are shown in Figure 5.

As shown in Figure 5, when the resulting nanosheets were exposed to a 100 mW 650 nm light resource, the photocurrent switching phenomenon was determined. The response time, recovery time, and on/off ratio of MoO_x_S_y_ to the 100 mW 650 nm light resource are shown in Table 1. Since 650 nm is close to 700 nm, the utilization of most of the visible light region is not expected to be a problem.

In biomedical and information sciences, NIR-I (808 nm) and NIR-II (980 and 1064 nm) are important light sources in photodetectors, condensed matter physics, bioimaging, light/matter interaction, etc. Meanwhile, the ratio of light of different wavelengths in the solar spectrum is generally as follows. The ultraviolet region is about 4–6%, the visible light region is about 46% or so, and the NIR region is about 48–49% or so. Therefore, studying the use of NIR is not only extremely important for the material design itself but also offers a good guide for interdisciplinary applications. The photocurrent responses of MoO_x_S_y_ to the NIR-I and NIR-II were also investigated. The representative results are shown in Figure 6, Figure 7 and Figure 8, respectively.

As shown in Figure 6, when the resulting nanosheets were exposed to a 200 mW 808 nm light source, the photocurrent switching phenomenon was observed. The response time, recovery time, and on/off ratio of MoO_x_S_y_ to the 200 mW 808 nm light resource are shown in Table 1. Here, 808 nm is one of the important light sources of the NIR-I region. It would have promise for important applications in interdisciplinary fields such as biomedical applications.

As shown in Figure 7, when the resulting nanosheets were exposed to a 200 mW 980 nm light source, the photocurrent switching phenomenon was measured. The response time, recovery time, and on/off ratio of MoO_x_S_y_ to the 200 mW 980 nm light source are shown in Table 1.

As shown in Figure 8, when the MoO_x_S_y_ nanosheets were exposed to a 40 mW 1064 nm light source, the photocurrent switching phenomenon was observed. The response time, recovery time, and on/off ratio of MoO_x_S_y_ to the 40 mW 1064 nm light source are shown in Table 1. The resulting nanosheets still showed a measurable photocurrent signal to the weak 1064 nm NIR. Here, 980 nm and 1064 nm are representative in the NIR-II region. Therefore, the resulting heterojunction has potential applications in the NIR-II region.

It can be seen from the above results that the photocurrent sensitivity of the prepared nanosheets at 650, 808, and 980 nm is much higher than that at 1064 nm. The MoO_x_S_y_ nanosheets showed good photocurrent signals with different wavelength light sources’ excitation. This is mainly due to the band gap modulation by surface doping. When the energy of the incident light is greater than the bandgap width, the electrons jump from the ground state to the excited state, and the photogenerated carriers transported under the built-in electric fields or an applied bias voltage are led out of the loop to generate a photocurrent. Here, the optoelectronic responses have been further investigated using the lower power of typical light sources. The representative results are shown in Figure 9, Figure 10 and Figure 11.

As shown in Figure 9, it is found that the MoO_x_S_y_ nanosheets still have a good photocurrent response to 50 mW 650 nm of light source. However, there is almost no optoelectronic response to 5 mW 650 nm of light. These results show that the photogenerated carriers are all trapped by the material defects when the energy excitation is less than 5 mW 650 nm of light. There is no net increase in the photoelectric signal.

Similarly, as shown in Figure 10, the resulting nanosheets still have a good optoelectronic response to 50 mW 980 nm of light. However, there is almost no optoelectronic response to 5 mW 980 nm of light. The MoO_x_S_y_ nanosheets therefore have a threshold value. If the power of the excitation light is below this critical value, the photogenerated carriers are all trapped by the defects in the material. The photocurrent could not be determined. Only when the power of the excitation light is above this critical value can the photogenerated free carriers be measured.

In short, the MoO_x_S_y_ nanosheets showed good photocurrent responses from the visible light region to the NIR. However, MoO_3_ is a wide band gap semiconductor material, where its band gap width is about 3.1 eV. The light above 400 nm is very difficult to absorb. The surface S-doping of MoO_3_ improved the optical absorbance and photocurrent in the NIR. As the exchange of O and S elements occurred on the surface of MoO_3_, it led to the formation of a MoO_3_/MoS_2_ surface heterojunction. At a certain depth and in a certain region of the MoO_3_/MoS_2_ surface heterojunction, there would be a region of surface S-doped MoO_3_ and surface O-doped MoS_2_. This region can be considered an interdoping of O and S. Due to the large difference between the atomic radii of sulfur and oxygen, the mutual doping of sulfur and oxygen elements also leads to changes in the band gap width of the internal stress modulation. Otherwise, as MoS_2_ is a narrow band gap semiconductor material, the resulting MoO_3_/MoS_2_ surface heterojunction would cover the visible and NIR regions of absorption. A schematic diagram of the interfacial interaction of the MoO_3_/MoS_2_ surface heterojunction is shown in Figure 11.

Combining the schematic interface interaction in Figure 11, it can be seen that under irradiation with a longer wavelength light source, the excitation and transfer of electrons mainly come from MoS_2_ due to its narrow band gap semiconductor. The strong interfacial interaction of the MoO_3_/MoS_2_ surface heterojunction plays an important role in the charge transport, which contributes to the photocurrent extraction from the visible light region to the NIR due to the integrated wide and narrow band gap semiconductor materials. Meanwhile, the surface S-doping of MoO_3_ also leads to the formation of an oxygen vacancy and sulfur vacancy of MoO_3−x_S_2−y_. MoO_3−x_ has a good plasmonic characteristic, which further promotes interfacial charge transfer by light induction.

The above photoelectric signals are obtained using gold gap electrodes and PET film substrates for MoO_3_ nanosheets with surface sulfurization for 6 h, which make it easy to obtain physical signals due to the low contact barrier of the gold electrodes. We also used polyimide film substrates and carbon electrodes to examine the photoelectric signals over a wide spectral range for material samples with different surface sulfurization times (6, 4, and 2 h). Similarly, good photoelectric signals were obtained. These are not shown here for reasons of space. Heterojunctions are formed by the sulfidation of the oxide surface, and the most important feature of heterojunctions is the formation of built-in electric fields. The built-in electric field can drive the transport of photogenerated carriers. Therefore, some resulting samples were selected to investigate the effect of the bias voltage on the photogenerated current. Representative results are shown in Figure 12.

As shown in Figure 12, a good photoelectric signal was still obtained even with zero bias applied. This indicates that the formation of the surface heterojunction generates a strong built-in electric field that drives the photogenerated carrier transport, which can be self-powered. This is shown in Figure 12A. It is also shown that the interfacial charge transfer of the heterojunction generates a built-in electric field, which not only promotes the separation and transport of photogenerated carriers but also effectively prevents the recombination of photogenerated carriers. As shown in Figure 12B, as the bias voltage increases, not only does the dark current increase but also the switching ratio increases significantly. This indicates that increased bias voltage would enhance the photogenerated carrier transport. The application of bias voltage increases the driving force for photogenerated carrier transport.

As a result of the broad-spectrum photoelectric signals from surface S-doping, to further investigate the effects of surface sulfurization on the composition of the heterojunction’s photophysical properties, Raman spectra and 2D Raman mapping were carried out. Although the Raman spectra show some characteristic peaks of MoS_2_ and MoO_3_ in the resulting product, 2D Raman mapping proved difficult to perform. Therefore, EDS and EDS mapping were investigated. The Raman spectra are shown in Figure 13. The EDS and EDS mapping are given in Table 2.

As shown in Figure 13, the MoO_3_/MoS_2_ surface heterojunction with different surface sulfurization times (4 and 6 h) showed clearly the broad characteristic peaks of MoS_2_ at 399.3 and 398.5 cm^−^^1^, which correspond to the A_1g_ vibrational modes of MoS_2_. The broad characteristic peaks of MoO_3_ at 814.9 and 815.3 cm^−^^1^ are assigned to the stretching modes of Mo_2_−O. The broadening of the Raman characteristic peaks of MoS_2_ and MoO_3_ may be due to a combination of ordered, disordered, and structures with some defects. When the 2D Raman mapping was measured, it was found that the characteristic signal peak would shift from molybdenum sulfide to molybdenum oxide when the resulting sample was irradiated by laser for a long time, so the Raman mapping is not very good. These results also suggest that Mo sulfides oxidize to oxides under the conditions of light irradiation. This also shows that oxides and sulfides can be transformed into each other by adjusting the experimental conditions.

As shown in Table 2, it is found that the sulfur content of the resulting product increased when increasing the surface sulfurization time and the oxygen content decreased with an increasing sulfurization time. The high oxygen content indicates that the resulting product is still dominated by Mo oxides. As shown in the EDS mapping of the MoO_3_/MoS_2_ heterojunction with different surface sulfurization times, the distribution of the S, Mo, and O elements is uniform in the MoO_3_/MoS_2_ heterojunction. It is shown that surface S-doping is feasible for oxide surface modification.

## 4. Conclusions

In summary, MoO_x_S_y_ nanosheets were obtained by the surface S-doping of oxide via a two-step hydrothermal process. The strong interfacial interaction of the MoO_3_/MoS_2_ surface heterojunction enhanced the photogenerated carrier extraction ability. The resulting MoO_x_S_y_ nanosheets exhibited broadband spectral photocurrent-extracting behaviors. Even with zero bias applied and carbon electrodes, a good photoelectric signal was still obtained. This indicates that the formation of the surface heterojunction generates a strong built-in electric field that drives the photogenerated carrier transport, which can be self-powered. The photocurrent sensitivity of the resulting nanosheets to 650, 808, and 980 nm is much higher than that to 1064 nm. Since the exchange of O and S elements on the surface of MoO_3_ was performed during the material synthesis process, it naturally led to the formation of an MoO_3_/MoS_2_ surface heterojunction. In some regions of the MoO_3_/MoS_2_ surface heterojunction, there would be the presence of the region of surface S-doped MoO_3_ and surface O-doped MoS_2_. This region can be considered an interdoping process of O and S elements, which led to the production of an oxygen vacancy and sulfur vacancy of MoO_3−x_S_2−y_. The plasmonic characteristics of MoO_3−x_ contribute to the interfacial charge transfer. This is a simple and inexpensive approach with which to obtain surface S-doped metal oxides with a good photocurrent response from the visible to the NIR region. This method can be extended to the surface functionalization of some similar oxide materials.

## Data Availability

The data presented in this study are available on request from the corresponding author. The data are not publicly available due to privacy.

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
