# Peer review of "Construction of a MoOx/MoS2 Heterojunction via the Surface Sulfurization of the Oxide and Its Photocurrent-Switching Characteristics in the Range of the Broadband Light Spectrum"

_materials, 2024, doi:10.3390/ma17225507_

Round 1
Reviewer 1 Report
Comments and Suggestions for Authors
See the attached file

Author Response
Dear Sir,
Thank you for your excellent question. We have revised it according to your comments. Major changes have been marked in the revision.
Sincerely,
Xingfa Ma

Reviewer 2 Report
Comments and Suggestions for Authors
Referee report on “Construction of Surface MoOx/MoS2 Heterojunction by Surface S/O Exchange of Oxide and its Photocurrent-switching Characteristics in the Range of Broadband Light Spectrum”
Although this is an interesting article, it cannot be recommended for publication because it needs some improvement.
1. The introduction is written in detail and in depth.
2. Lines 199-201. There is no information on the purity of the samples. It would be useful to indicate the corresponding websites.
3. Scheme 1. The top inscription is difficult to see.
4. Scheme 2. The quality of the drawing is poor, as if it was made with glue and scissors from other drawings.
5. Figure 1. The homogeneity claim would be useful to confirm by doing a 2D Raman mapping.
6. Lines 250-263. This sentence “some diffraction peaks still can be distinguished” is the author's imagination, in reality there is only noise here.
7. Line 264. “It shows a clear absorption peak in this range”. If so, please provide its values (peak position and bandwidth in eV) ​​and analyze the data obtained. For the reviewer, the presence of this peak is not obvious.
8. Figure 4. What is the wavelength of the visible light?
9. Figures 5-8. It is unclear how light of this wavelength creates carriers.
10. Table 1. Given the noise level (Fig. 5-8), the values ​​presented here are given with excessive precision, the measurement error is not given.
Author Response

(The authors gave the same response as above.)

Reviewer 3 Report
Comments and Suggestions for Authors
The work presented by Ma et al is not ready to be submitted to be consider for publication neither in the journal Materials nor any other in the current form. Therefore, I kindly ask the authors to rewrite it and resubmit it.
-In an original scientific work article, an introduction section cannot have a review of the literature in the field. 101 references is an excessive number. Please, reduce the number of references to keep just the ones that are related closely to the work developed in the manuscript, and those that are important to reinforce the novelty and the motivation of the work compared to the previous similar works in the field.
-In the introduction section, the authors should explain more in detail the novelty, the relevance and the motivation of this work compared to previous works in the field.
-The abstract is too long and general
-The section: "2. Experimental Part" should be developed, written with more details. The authors cannot just send the readers the synthesis steps, materials and tools details, measurement parameters...to read previous works from the authors.
-Which is the value of the thickness and the gap distance of the Au gap electrodes?
-In figure 2, please, mark the diffraction peaks on the graph for the readers
-In line 264: "The absorbance strength in the 700-800 nm range was a little high" But in figure 3, the absorbance in the range of 300-400 nm is higher. The author did not mention/explain about this results.
-The authors wrote: "As shown in Fig. 4, when the MoOxSy film was excited with the weak visible light, the current increased dramatically. On the contrary, the film current decreased significantly when the visible light was off." But in figure 4, the ON and OFF states are written in the other way. Which option is the correct? Are in Figures 5-8, the ON and OFF states also correct?
-The results and discussion from the characterization measurements should be analysed and explain more in depth
-The provided model to explain the mechanisms of photodetection due to the formation of the heterojunction should be developed in more detail.
Comments on the Quality of English LanguageThe option "Extensive editing of English language required" was selected because the authors have to rewrite significantly the manuscript. There are many sentences that are not well written grammatically. There are typos and mistakes. There are sentences that do not follow the formal style expected in a scientific publication. Since this affect the whole manuscript, I would not give the full list, but I would provide some examples that need amendment:
-In lines 155 and 159, the authors wrote MnO2 instead of MoO3
-In lines 243-245, the sentences: "As shown in Fig.1, the whole morphology was emerged nanosheet structure, its length is about 100 nm or so." and "The thickness is 5-6 nm or so."The first sentence is gramatically not well written and both sentences are too informal to be used in scientific writing
-In line 254: "or it is maybe the result of a thinner film of MoOxSy for XRD determination." This sentence is grammatically wrong
Author Response

(The authors gave the same response as above.)

Reviewer 4 Report
Comments and Suggestions for Authors
Review of the manuscript Materials-3232124 for the Authors: This manuscript presents a study on the construction of surface MoOx/MoS2 heterojunctions. The research is sufficiently novel, and the manuscript is generally well-written. However, several issues need to be addressed, particularly in the results section, before the manuscript can be considered for publication. Please see the following detailed comments.
1) Title – The title could be shortened. While specificity is important, it is currently too long and can be more concise. Abstract – Similarly, the abstract is too lengthy. It would benefit from briefly summarizing the system, key results, novelty, and significance, while avoiding unnecessary detail.
2) Avoid grouping too many references together in the Introduction (e.g., 8 to 20). This is not a review paper, so such extensive referencing is unnecessary. The language and flow need improvement in places. For instance, the sentence: “Therefore, MoO3 and MoS2 can be transformed by each other via control process technology and surface treatment factors, resulting in changes to the photophysical properties of the materials in the visible region and part of the NIR” is unclear and should be rewritten. Additionally, the introduction is too long and should be more concise; appropriate for a research paper rather than a review.
3) Experimental Part – This section is generally acceptable, but more details should be provided rather than simply directing readers to references. For example, instead of: “The detail of characterizations of SEM (scanning electron microscope), UV-Vis-NIR (UV-VIS-NIR spectrophotometer), and XRD (X-ray powder diffraction) was similar to the reference [101],” the methods should be described more explicitly.
4) Scheme 1 – Improve the resolution, and consider using a clearer font. Scheme 2: Improve the resolution.
5) Figure 1: Double-check the scale bars. Are both images truly at 200 nm scale?
6) The XRD data in its current form appears to be noise. Please improve the scan and present more reliable data. Optoelectronic analysis: This section is generally acceptable.
7) Figure 11: This figure needs significant improvement in quality.
8) Conclusions – This section is well-written and can remain as is. Literature – Adequate.
Therefore, my current recommendation is major revision.
Comments on the Quality of English LanguageThe language is mainly ok.
Author Response

(The authors gave the same response as above.)

Round 2
Reviewer 2 Report
Comments and Suggestions for Authors
After successful revision, this manuscript can be recommended for publication.
Author Response
Dear Sir,
Thank you for your work and your comments. We have revised it according to the comments. In some places a reasonable explanation was given. Major modifications are identified.
Sincerely,
Xingfa Ma
After successful revision, this manuscript can be recommended for publication.
Thank you for your work and your excellent question.
Reviewer 3 Report
Comments and Suggestions for Authors
The authors have not amended fully all the points I have kindly asked for and they have tried to make as small modifications as possible in their original submitted manuscript, which was highly recommendable. However, they have improved the manuscript adding more relevant experimental data and analysis of results and they have reduced a significant amount of references as requested.
Comments on the Quality of English LanguageThe authors have improved the manuscript
Author Response
Dear Sir,
Thank you for your work and your comments. We have revised it according to the comments. In some places a reasonable explanation was given. Major modifications are identified.
Sincerely,
Xingfa Ma

Reviewer 4 Report
Comments and Suggestions for Authors
Review of the manuscript Materials-3232124 v2 for the Authors: This is the revised version of the manuscript presenting surface MoOx/MoS2 heterojunctions study. In this revised manuscript, the authors have addressed many of the questions raised in the initial round of reviews. However, some areas still require attention. Please see the following comments:
1) The introduction remains overly lengthy, with too many references, some of which are still presented in large groups. It would be beneficial to shorten this section and streamline the references for improved clarity.
2) The experimental section is now clearer and much improved. The results section shows significant improvements. However, I have a question regarding the XRD data: was the observed low crystallinity expected for a MoOx/MoSâ‚‚ system, or not? Please consider adding references to support this observation to confirm your findings.
The language quality has improved, but I recommend addressing these last few issues before publication. My recommendation is therefore a minor review.
Comments on the Quality of English Language
Much better.
Author Response

(The authors gave the same response as above.)
